# Use of LiDAR Technology for the Study and Analysis of Construction Phases and Deformations in the Gothic Church of Biar (Spain)

Jose Antonio Huesca-Tortosa *, Yolanda Spairani-Berrio ⬚ and Pascual Saura-Gómez ⬚

Department of Architectural Constructions, University of Alicante, Carretera de San Vicente del Raspeig, s/n, 03690 San Vicente del Raspeig, Spain; yolanda.spairani@ua.es (Y.S.-B.); pascual.saura@ua.es (P.S.-G.)
* Correspondence: ja.huesca@ua.es; Tel.: +34-965903400 (ext. 2020)

**Abstract:** This work provides new data on the church of Nuestra Señora de la Asunción in Biar, combining the investigation of documentary sources with the analysis of the geometry obtained using LiDAR technology and the study of stone materials. This monument has a Proto-Renaissance façade of great architectural value, as it was the first building in the province of Alicante, in Spain, to use Renaissance decorations at the beginning of the 16th century. Its main façade reflects the four centuries of its construction from the 15th to the 18th centuries. The building has been digitised using LiDAR technology and photogrammetry. The graphic representation of the point cloud obtained from the aforementioned techniques makes it possible to study deformations in colour gradient with respect to a reference plane. The results obtained after the analysis of these data show that the method used in this work has served to detect and corroborate the constructive evolutions of the church obtained from the documentary sources investigated. This work serves as an example for similar works, proposing the incorporation of the analysis of anomalies in the geometry of the facings as a new variable that should be considered to complement the rest of the usual studies, bringing to light deformations and irregularities that at first sight may go unnoticed.

**Keywords:** 3D laser scanning; photogrammetry; deformation control; proto-renaissance; cultural heritage; LiDAR

## 1. Introduction

The church of Nuestra Señora de la Asunción is located in the centre of Biar (Alicante) in the southeast of Spain. It is listed together with its immediate surroundings as a Monument of Cultural Interest (BIC: Bien de Interés Cultural). Its architectural and constructive typology can be classified as Late Gothic, as its interior structure is formed by stone vaults in the form of ribbed vaults separated by pointed arches.

The Proto-Renaissance façade on the northeast-facing façade is of great value for being the first building in the province of Alicante to contain decorative and formal elements typical of the Renaissance.

The construction of the current building began in the 15th century, although the exact date is unknown, and was completed at the end of the 18th century. The architecture and the different construction techniques of the church reflect its lengthy construction.

The aim of this work is to understand the constructive evolution of the building by interweaving the documentary data with the data obtained from the analysis of its geometry (point cloud) and its materials. The study focuses on the northeast façade, which corresponds to the main façade of the building. This façade contains walls built in all the construction stages that have shaped this monument, as well as the Proto-Renaissance façade.

Until now, hardly any studies had been carried out on this building [1–5], which is why this work provides new data on its constructive evolution and its current state of conservation. The study has been based, on the one hand, on the research, compilation and

analysis of written documentation, and on the other hand, on the in situ reconnaissance of the building, analysis of the masonry and its materials, and the comparative study of the deformations and anomalies obtained by means of point clouds, applying LiDAR (Light Detection And Ranging) technology and photogrammetry (Figure 1).

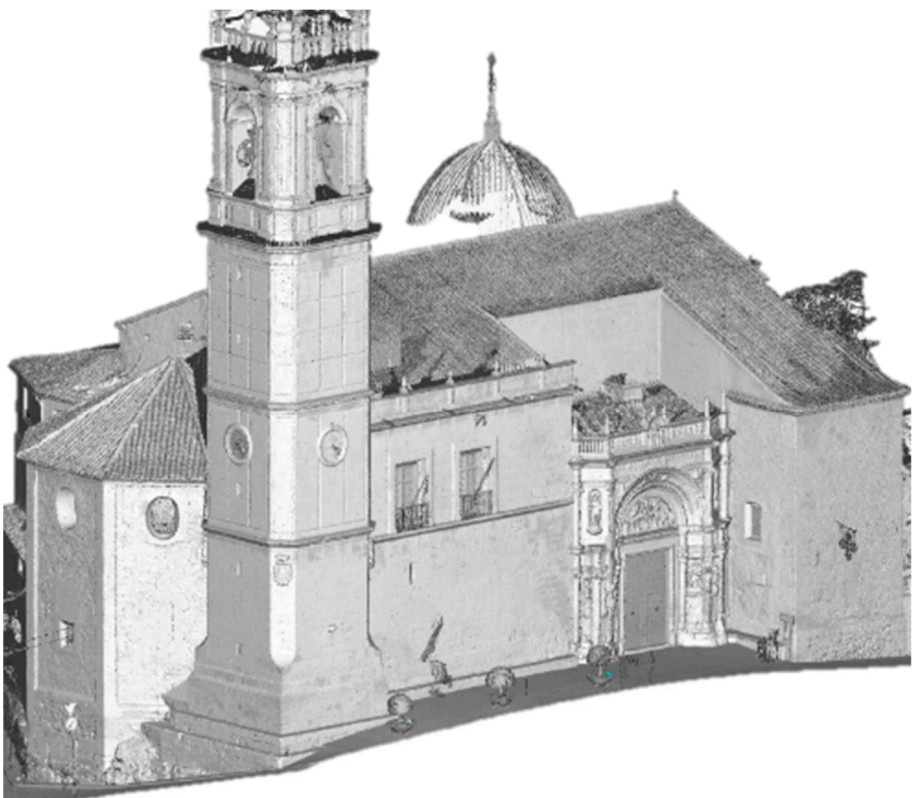

**Figure 1.** Modelling of the church based on a point cloud obtained with LiDAR. Northeast façade containing the Proto-Renaissance doorway.

The use of LiDAR technology for the recording and diagnosis of historic buildings has been established for many years, as demonstrated by numerous authors who have worked on the advancement of these techniques [6–12]. However, each building is unique and poses different challenges.

Obtaining precise measurements using these techniques has also made it possible to establish the reference units and modules used in the construction of the building, significant data that can be used to determine different periods of construction.

*1.1. Historic Background of the Building*

The origin of the current building dates back to the time of the Christian conquest in 1245 by Jaime I [13]. Until that time, the town of Biar was populated by Muslims [14], and in part of the floor that the church occupies today there was a mosque that, when the town was conquered, would become Christianised, as was customary in those cases [15]. One of the first documents that speak of the first church dedicated to Santa Maria is the will of Llorenç d'Escala dated 17 December 1305, according to López Albero [2] (information extracted from the Archive of the Kingdom of Valencia, Manaments i empares, year 1682, lib. 3).

This building could correspond to the mosque mentioned above, which had been Christianised and was located next to the wall. Thanks to the research carried out by the authors of this work, the "Protocolos Notariales de Antonio Aranda" (Notarial Protocols of Antonio Aranda) were located in the Municipal Historical Archive of Biar, in which there are references to payments for works in the church from 1490 [16] to 1496 [17]. In 1493 [18]

and 1496, payments are described for the execution and finishing of the side chapels. Each chapel is assigned to a saint or saints, and the documents consulted coincide with the current dedications of these chapels. This dates the completion of the three naves of the church to the end of the 15th century.

The Proto-Renaissance doorway (Figure 2a) was built between 1519 [19] and 1522 [20], attached to the bell tower (which already existed). From the construction of the façade until the end of the 17th century, the documents consulted do not show any work on the building.

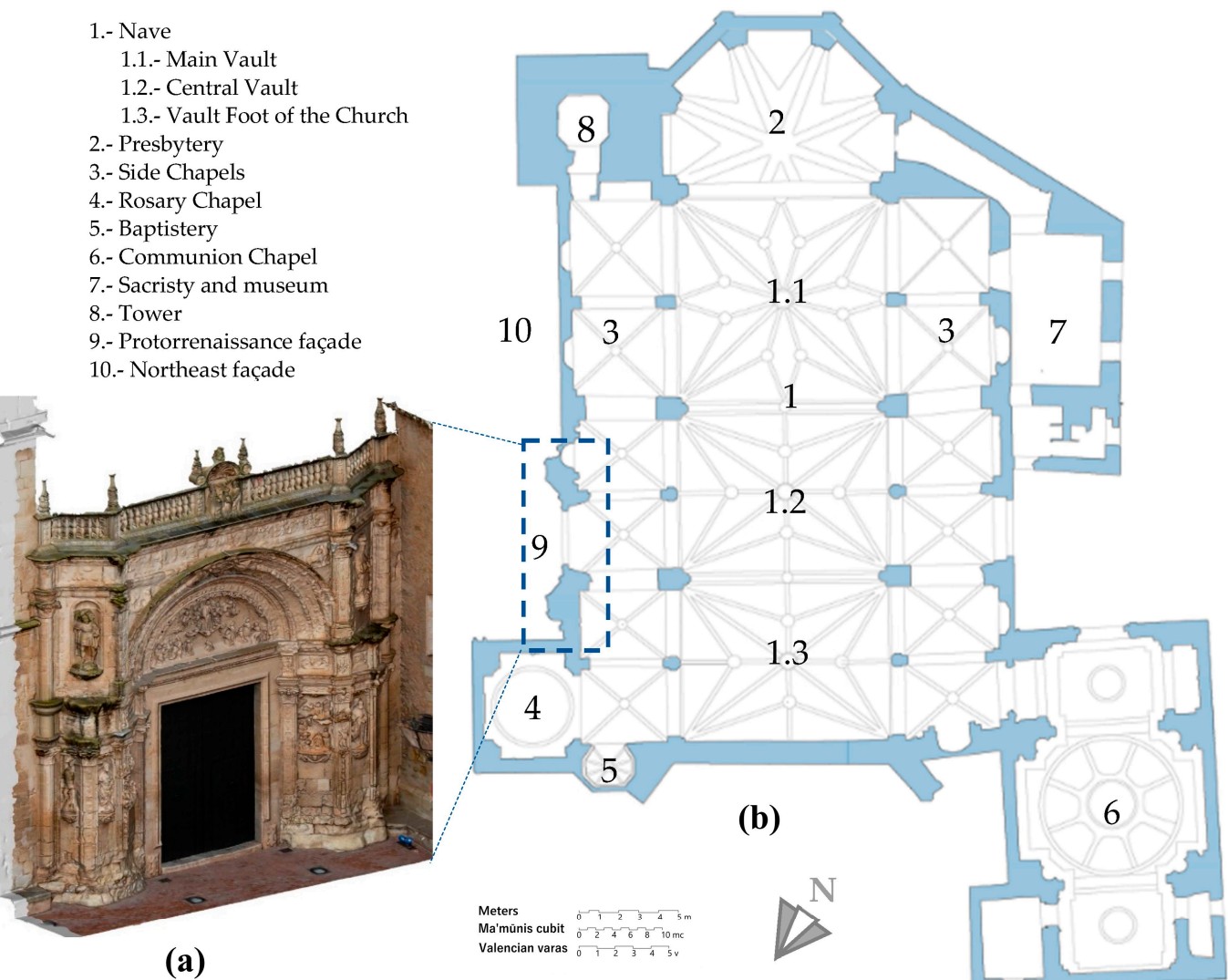

1.- Nave
   1.1.- Main Vault
   1.2.- Central Vault
   1.3.- Vault Foot of the Church
2.- Presbytery
3.- Side Chapels
4.- Rosary Chapel
5.- Baptistery
6.- Communion Chapel
7.- Sacristy and museum
8.- Tower
9.- Protorrenaissance façade
10.- Northeast façade

Meters
Ma'mūnis cubit
Valencian varas

**Figure 2.** (**a**) Proto-Renaissance doorway. (**b**) Plant with identified parts.

At the end of the 17th century, the Communion Chapel was built at the foot of the church (indicated as 6 in Figure 2). The Communion Chapel, one of the most beautiful pieces of Valencian Baroque, was designed by Juan Bautista Pérez Castiel in 1688, executed by his brother Vicente Pérez Castiel, Antonio Pons and Bertomeu Mir, and completed in 1694 [21].

On 7 May 1699, the laying of the first stone of the Tower is documented. It seems that work really began in 1702 [4], as is attested by the lapidary inscription on the tower, which indicates that it was finished in 1767. Subsequently, the presbytery was also built in the 18th century.

The chapter house was located above the side chapels facing the northeast façade. Access to this space is from the stairs of the new tower. This room must therefore have been

built later than the tower. Furthermore, the architecture and construction systems used in the chapter house, as well as the interior decoration and the original Valencian-style glazed ceramic floor, which still exists, place this building in the 18th century.

Regarding the Chapel of the Rosary, there is uncertainty about its year of construction and how it evolved. Observation in situ and the architectural forms indicate that it was built attached to the Proto-Renaissance façade. The date of these works is not clear, although according to Payá [5] they were carried out in the 18th century.

### 1.2. Constructive Description of the Church

As mentioned above, the oldest part corresponds to the three naves of the church, the central one consisting of three sections covered by cross vaults separated by pointed arches. The vaults are built with a Gothic structure which, according to Ungewitter [22], derives from the combination of the construction procedures of the groin vault and the dome, seeking greater ease of construction and better stability. The pointed arches used between vaults are supported on slender pillars of reduced section, which give verticality to the Gothic design and resolve the structure with fewer thrusts than the round or segmental arches transmitted in the previous constructive solutions [23].

The vault located further south (numbered 1.1 in Figure 3) is made of carved stone with ribs in the shape of an eight-pointed star and, in plan, corresponds to the proportion "ad quadratum" [24–26]; its side measures 10.08 m, which does not belong to the Valencian Gothic, according to Zaragoza [26]. This figure corresponds exactly to 24 cubits Ma'mūnis, the usual measurement in the Almohad period, which corresponds to 42 cm [27,28]. The other two vaults have a rectangular plan, the proportions of which are obtained ad triangulum from the 24 cubits Ma'mūnis, and they are built with a vault with stone ribs forming tercelets. In the aisles, there are six ribbed vaults on each side.

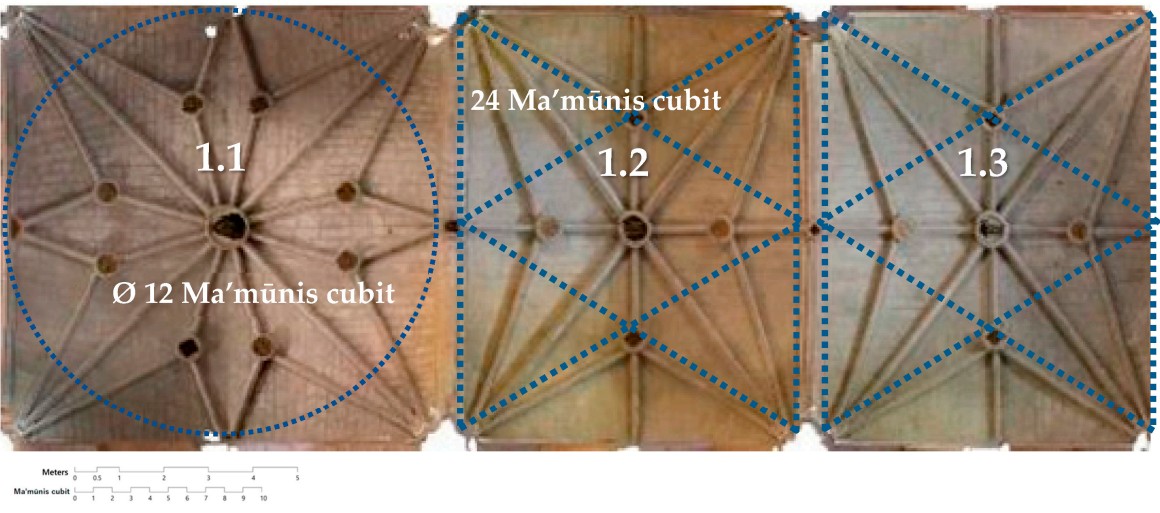

**Figure 3.** Orthophotography of the central nave vaults.

From the southwest chapel (next to the chancel) there is access to the sacristy, the parish quarters and the church archives (numbered 7 in Figure 2b).

The Communion Chapel has a Greek cross plan with short arms, covered by an octagonal dome on pendentives. It is profusely decorated with plasterwork in the form of acanthus and angels that cover the walls, vaults, pendentives and dome.

## 2. Materials and Methods

The methodology recommended by experts [29] and heritage protection organisations, such as ICOMOS [30], was followed for this work. In addition, as much written and graphic information as possible about the building has been collected from different archives, as well as personal interviews with people who have lived in the building and/or have

worked on it. In situ data were taken manually by means of sketches and instrumental techniques. The data obtained were processed and analysed.

The results have been used to contrast the previous hypotheses and/or to complete other studies, such as that of the materials used to build the different walls of the façade under study.

### 2.1. Compilation and Analysis of Existing Documentation

Existing information on the building has been compiled, and both written and graphic documents were found in the municipal archives of Biar and Villena, in the Chapter Archive of the Cathedral of Cordoba, the provincial historical archive of Cordoba, the parish archive of Biar, the church archive of Biar, the archive of the Provincial Council of Alicante, the Corpus Christi Archive of Valencia, the Archaeological Museum of Cordoba and the Ethnographic Museum of Biar.

### 2.2. Methodology In Situ Data Collection with Instrumental Techniques

LiDAR and photogrammetry techniques serve to digitize heritage reliably and accurately [6,31]. Orthophotos obtained from the 3D cloud point are usually used to document historical buildings, as currently presented. With these techniques there can also be obtained deformation and degradation with high precision in millimetre values on a cloud point with more than 540 thousand points (Figure 4) [7–12].

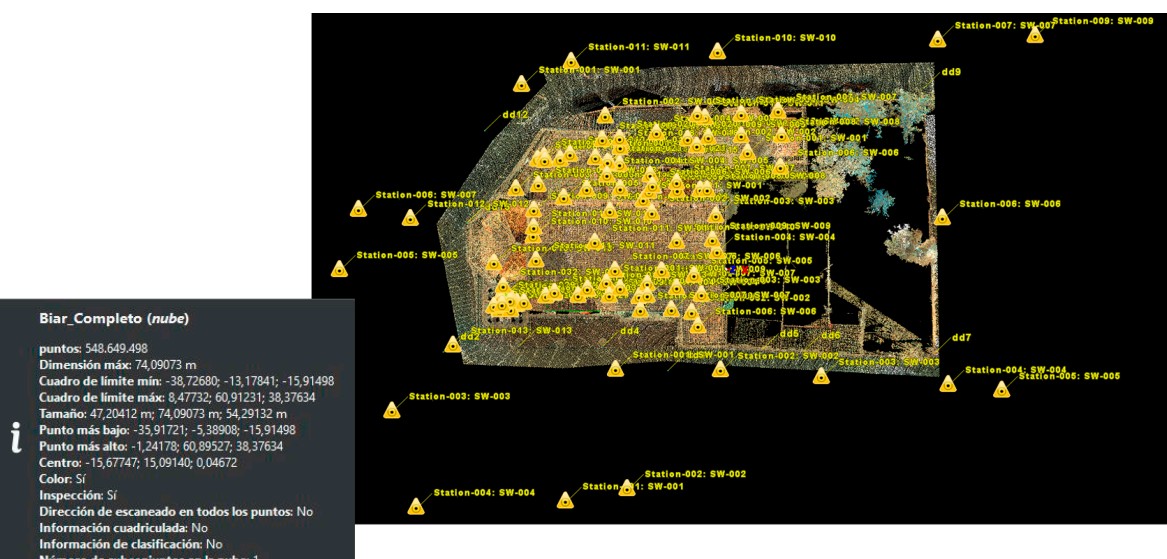

**Figure 4.** Location of all station points and point cloud volume.

Various tools were used for the 3D geometrical survey, such as the Leica Terrestrial Laser Scanners (TLS) model BLK to obtain 360 HDR images of the interior of the church and model C10 with their corresponding integrated digital cameras, as well as the software for their registration, processing and analysis of the point clouds.

Both digital images using photogrammetric techniques and captures with TLS itself obtain point clouds with their real colour using specific software (Metashape by Agisoft and Cyclone Register 360). The photos were taken with a Canon model 80D digital SLR camera with 24.3 MP CMOS APS-C sensor and 7.560-pixel RGB + IR metering sensor with an ISO sensitivity range of 100–16,000, using graphic reference scales, and also a DJI drone model Phantom 3 Pro with 1/2.3″ CMOS sensor 12.4 million effective pixels, GPS + GLONASS, and lens FOV: 94° 20 mm equivalent format: 35 mm and aperture of f/2.8 at ∞.

A preliminary study was carried out to plan the implementation and positioning of the apparatus, since the final result of the data collection process in the field depends on it. Mobile targets and fixed reference points were used, supported by buildings and adjacent

fixed objects, in order to be able to relate clouds to each other until a single complete cloud was obtained, accurately and with controlled measurement errors. More than 80 station points were carried out for the survey of the church (Figure 4). In this case, two image capture systems were used, one from digital devices without built-in geo-localisation, such as conventional digital cameras, and the other from a drone with a built-in geo-positioning system, using GPS + GLONASS.

### 2.3. Methodology in the Study of Material Samples

In order to know the different lithologies that make up this façade and to be able to corroborate the different construction stages, the stone materials of each area have been studied.

Following the recommendations of the Spanish Institute of Cultural Heritage (IPCE) [32,33], samples were taken from detached areas, doing as little damage as possible to the monument. A total of 4 samples of less than 10 cm$^3$ were taken. Part of each sample (approximately 1 g) was pulverised for XRD analysis.

Petrographic characterization of the samples was performed using a petrographic polarizing microscope (POM) and a Zeiss Axioskop transmitted-light optical microscope with a Zeiss Stemi SV 6 magnifying glass and a Photometrics Cool SNAP-CF camera. A thin section of stone was observed.

The Bruker D8-Advance equipment with a Göebel mirror was used for X-ray diffraction (XRD). All this equipment is part of the equipment of the Technical Services of the University of Alicante

### 2.4. Methodology in Analysing the Results Obtained

The original written documents found in this research were transcribed and reviewed by palaeographers from the University of Alicante. The results were analysed and the constructive evolution of the building has been proposed.

The clouds were downloaded one by one. Leica Cyclone software (version 9.4.2) was used for the global registration and compilation into a single total point cloud, with which the manual and automatic cloud-by-cloud alignment and registration was carried out, and the possible errors in the overlapping of clouds were inspected. The value of the absolute error in the alignment of all the records obtained from all the positions of the terrestrial laser scanner around the building is 5 mm.

The results of the geometric recording of the building were analysed and the data interlinked with the data obtained from documentary sources, with the results of the different lithologies of the factories studied and with the in situ observation of the construction elements (Figure 5).

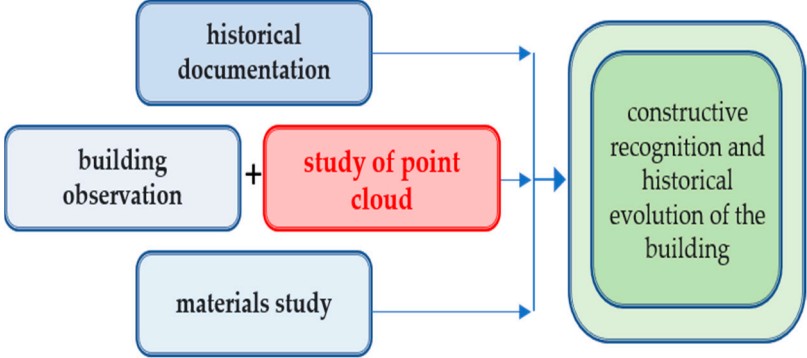

**Figure 5.** Workflow at this job.

### 3. Results

This section includes the results of the study of the geometry of the northeast façade and the characterisation of the materials that make it up.

The façade shows numerous constructive anomalies because of the sum of several masonries with different types of ashlars in terms of format and size. There are vertical and horizontal constructive joints typical of the construction of masonry in different stages of construction. This elevation of the building also shows numerous degradations due to the alteration of the stone materials of which it is composed. The areas affected by damp are the most altered. Some ashlars show a high loss of their original shape with multiple cavities, mainly due to alveolisation and sandblasting [34]. The most affected parts are the plinth in the lower part of the wall and the areas of incidence of rainwater runoff from the gargoyles on the vertical of the wall below its location (Figure 6).

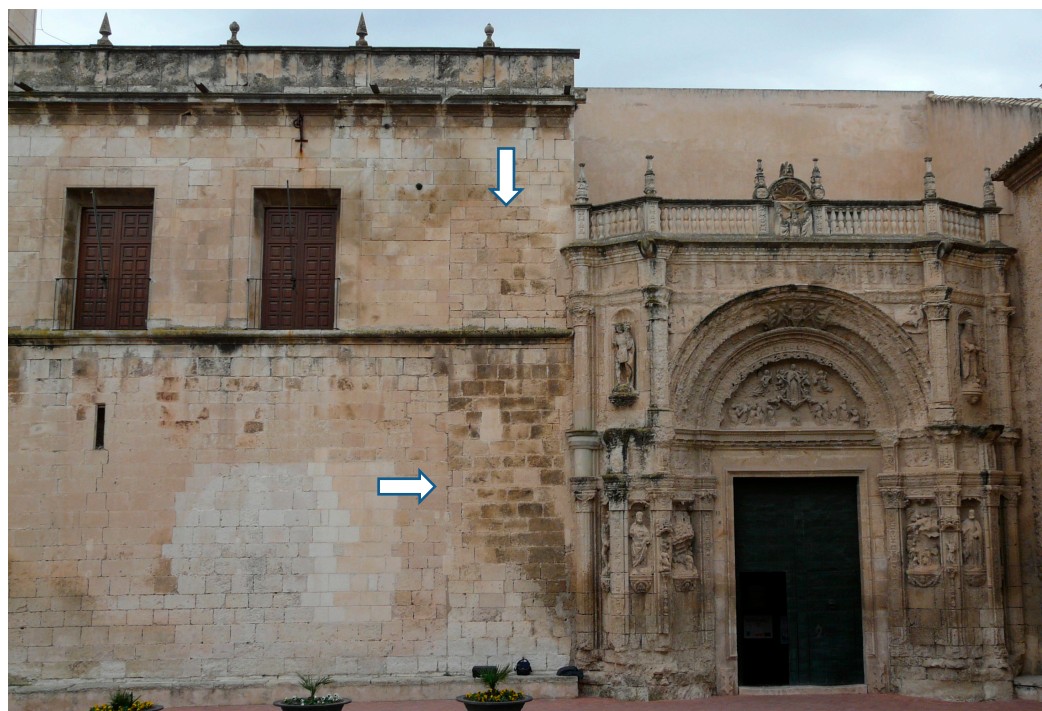

**Figure 6.** Northeast façade of the church.

### 3.1. Study of Anomalies on the Northeastern Façade

As we have seen above, the Proto-Renaissance doorway and the Late Renaissance chapels located to the northeast of the building were built attached to a masonry belonging to an old tower. Figure 6 shows the perfectly rectangular shape of the walls coinciding with the structure of that tower. This wall is the oldest construction element of the present building and therefore of this façade.

This tower was built with three leaves, two outer leaves of ashlar masonry with a masonry infill of small-calibre masonry and rubble in the core. These data were observed thanks to the existing cavities in the wall from inside the church on the upper floor. The thickness of this wall, as measured by LiDAR, is 85 cm plus its interior cladding, which corresponds to 2 cubits ma'mūnis.

The Late Gothic façade, consisting of the walls of the old tower and the enclosure of the Gothic chapels, contains 49 stonemason marks typical of the 14th–15th centuries and which only appear in this area of the building (Figure 7). Previously, other studies had been carried out on Gothic buildings in the surrounding area, such as the church of the Assumption in Villajoyosa [34] and the church of San Bartolomé in Jávea [35], and in the latter, coincidences were found with some of the stonemason marks in the church of Biar (Figure 7b).

The ashlars of this part of the main façade were carefully observed and a row was detected, very altered with large ashlars, marked in (Figure 7a), which could have been the

finishing touch of the façade of the chapels from the end of the 15th century until the 18th century, when the upper floor was built.

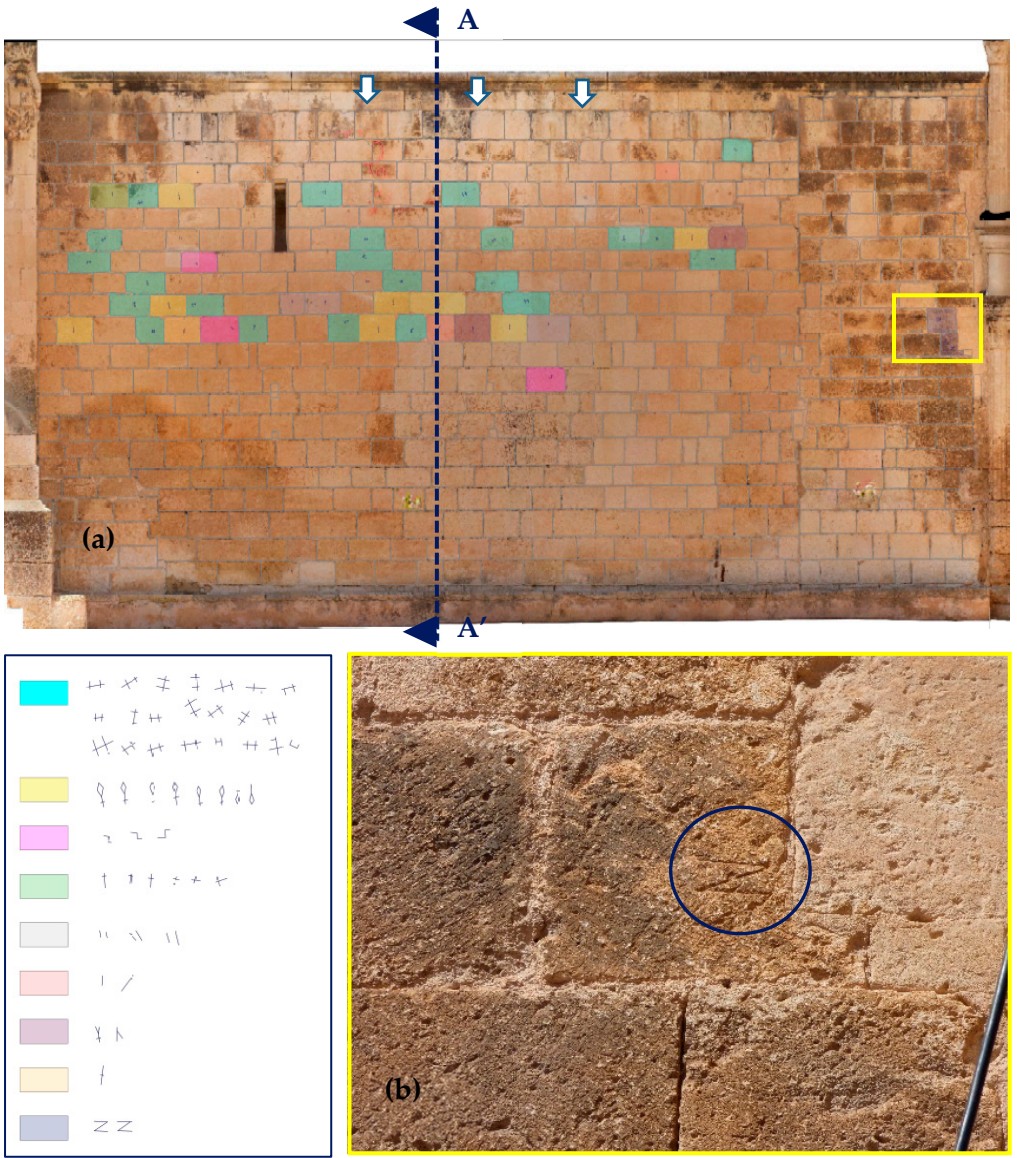

**Figure 7.** (**a**) Northeast façade of the church with ashlars coloured according to the type of stonemason marks, in the attached table. (**b**) Inside the circle detail of a mark that coincides with those of the Gothic church of Jávea (Alicante).

There is a Gothic rose window (highlighted in red in Figure 8) which does not currently bring light into the interior of the church, as it is partitioned off and faces the chapter house. Originally, this circular window would have been located in a wall to the outside.

The current roof overlooking this façade was built at the end of the 18th century when the Chapter House was built.

These data, together with the documentary studies of the construction evolution, have allowed us to hypothesis that the roof of this façade had, from its origin in the 15th century until the 18th century, only corresponded to the masonry that encloses the chapels.

In order to corroborate the above hypotheses, the deformations of this façade were analysed graphically by studying the point cloud obtained with LiDAR. All the analysis was carried out through the combined cloud of all the station point clouds obtained from a single Terrestrial laser scanner, the C10 model; specifically, the error of this station point is 4 mm and the total cloud-to-cloud error is 3 mm.

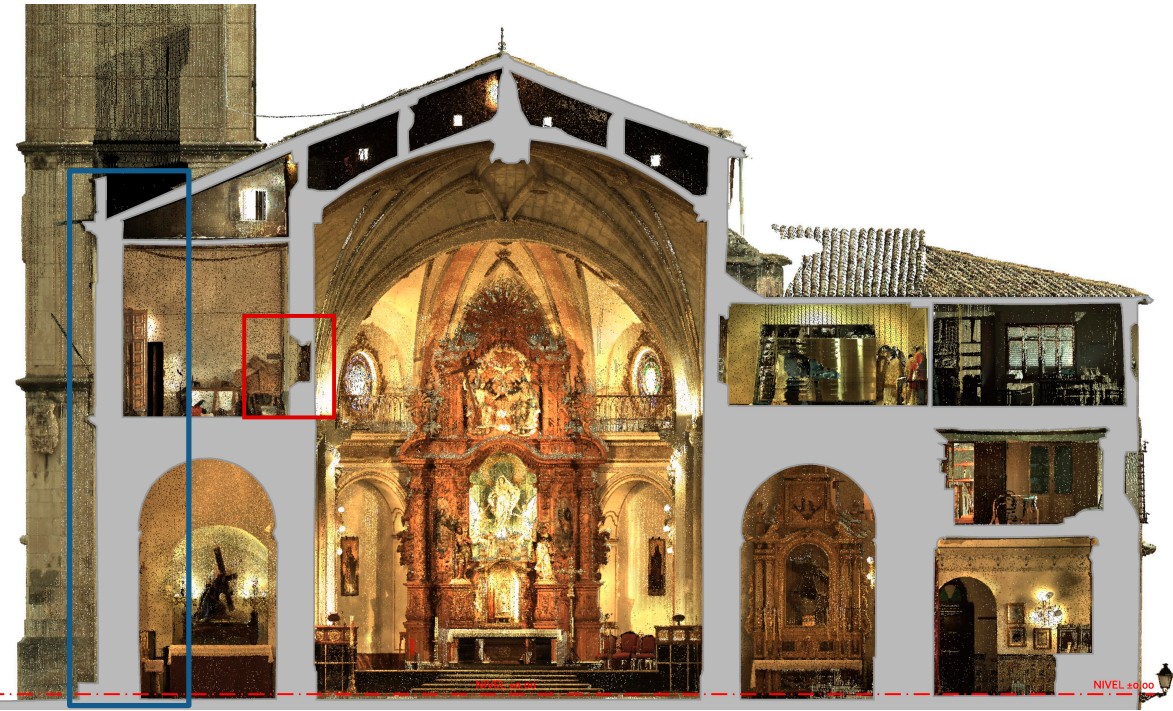

**Figure 8.** Church section, indicated in Figures 7a and 9c. The northeast façade under study is located inside the blue box. The Gothic rose window is located inside the red box.

The reference plane taken corresponds to the wall built at the end of the 15th century and is formed by three points on the façade of the chapels coinciding with the best possible plane of the wall without deformation. Subsequently, the deformations with respect to this plane were studied with different distance ranges.

In the different images in Figure 9, the three points numbered "69, 70 and 71" that form the reference plane are indicated with their coordinates. The green points coincide with the plane. The red and blue colours indicate the distance from the reference plane (red towards the inside of the building and blue towards the outside). Surveys were carried out with three different ranges from (+10 mm) to (−10 mm) away from the plane (Figure 9a), (+20 mm) to (−20 mm) away from the plane (Figure 9b) and (+30 mm) to (−30 mm) away from the plane (Figure 9c).

The areas shown in black in the figures are located outside the study range. If the part closest to the black area has red borders, it means that the black area is behind the reference plane. If the black area has blue borders, the black part is outside the building and ahead of the reference plane, e.g., the part indicated as "6" in Figure 9c.

It has been observed that the central area of the façade protrudes outwards by more than 3 cm. This part coincides with the horizontal thrust of the arch between the chapels that absorbs the loads of the vaults and transmits them to the wall, indicated as "6" in Figure 9c.

The row of ashlars below the current intermediate cornice, indicated as "5" in Figure 9c, is not built on the same plane as the lower masonry and corresponds to the Gothic façade. In addition, the row of ashlars just below shows much greater degradation, as indicated above. This could indicate that the Late Gothic masonry was finished off in the deteriorated course with larger ashlars and therefore the Baroque masonry begins in the course indicated as "5".

The plane that forms the façade of the original tower shows similar gradients of colours in vertical strips, which indicates that it is rotated with respect to the reference plane. The alignment of the plane of the chapel enclosure does not coincide exactly with the alignment of the former tower's masonry, although it may appear so to the naked eye.

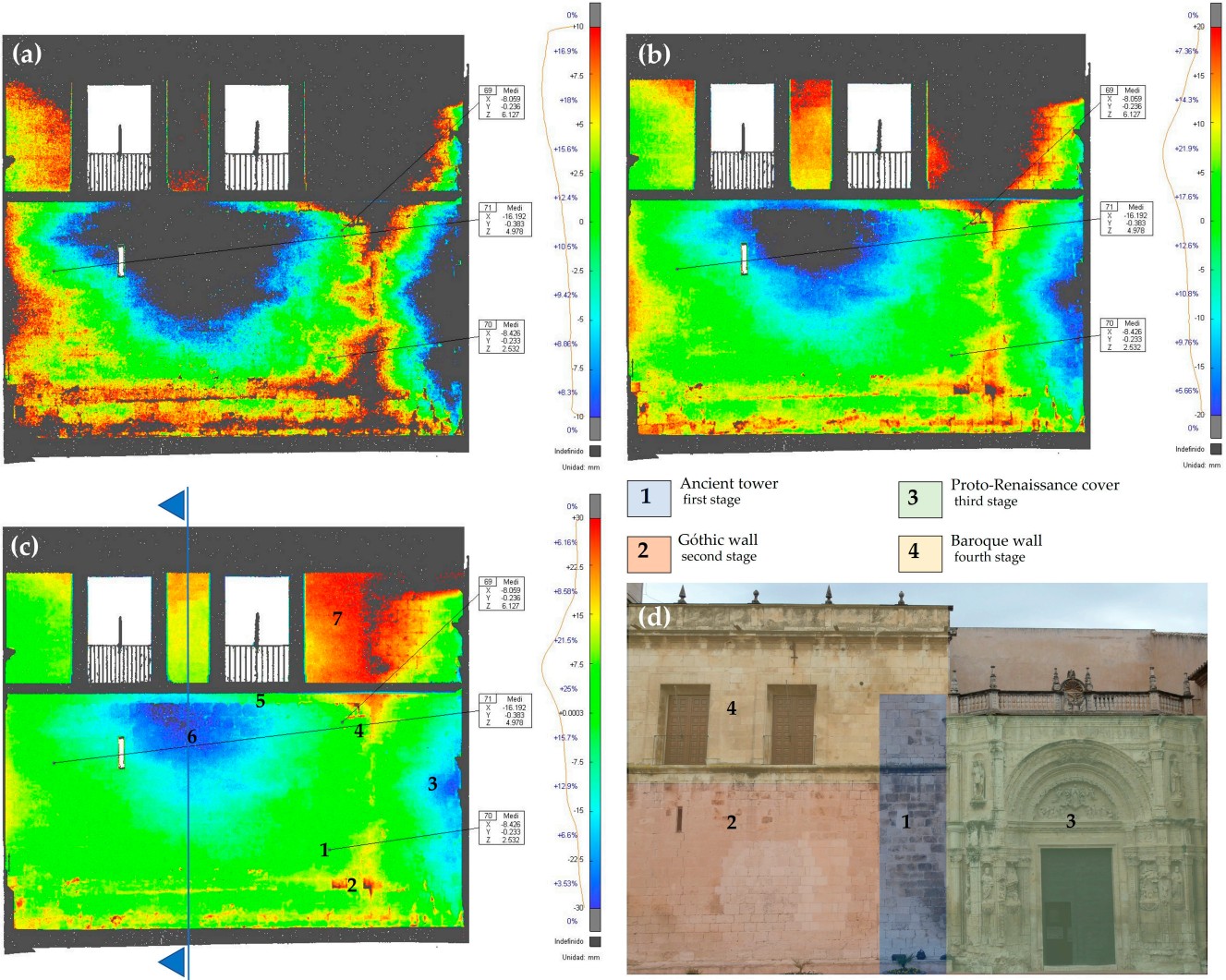

**Figure 9.** (**a**–**c**) Study of deformations in the façade in colour gradient. The green dots indicate the reference plane, the red and blue dots move away 10 mm in case (**a**), 20 mm in case (**b**) and 30 mm in case (**c**). The blue areas move away from the building, indicating outward slumps. The black areas of the wall do not fall within the range under study and are greater than +/−10mm, +/−20mm and +/−30 mm in figures (**a**–**c**). (**d**) Image of different construction stages.

At the point where it meets the doorway, there are anomalies that may correspond to the ashlars of the cornices of the Proto-Renaissance doorway being in contact with the pre-existing masonry of the tower.

In the upper part of the old tower, coinciding with the wall that was rebuilt in the Baroque period, there are deformations in deep red (indicated as "7" in Figure 9c) and black areas, indicating that there are parts of this wall that go into the interior of the building with respect to the reference plane.

As for the degradations of the walls, the lower part of the façade shows multiple anomalies, as well as the encounter with the masonry belonging to the original tower, indicated as "2 and 4" in Figure 9c.

### 3.2. Study of Stone Materials

The stonework of the different parts of the façade has been characterised with a double purpose, on the one hand to know the materials that make up the building and on the other to corroborate the possible constructive evolutions. They are described below in the order of the building's constructive evolution.

### 3.2.1. Ancient Tower Stone (s. XIV?-XV)

It is a white fine-grained biocalcarenite with high matrix content (biomicrite limestone). Fossil content corresponds to foraminifera and fragments of shells and echinoderm plates (according to POM observations) (Figure 10). Terrigenous content is low (<3%) and corresponds mainly to quartz, with trace contents of feldspar, opaque (dendritic structures) and clay minerals. According to XRD analysis, the mineralogical composition of this rock consisted of 95% calcite and 5% quartz.

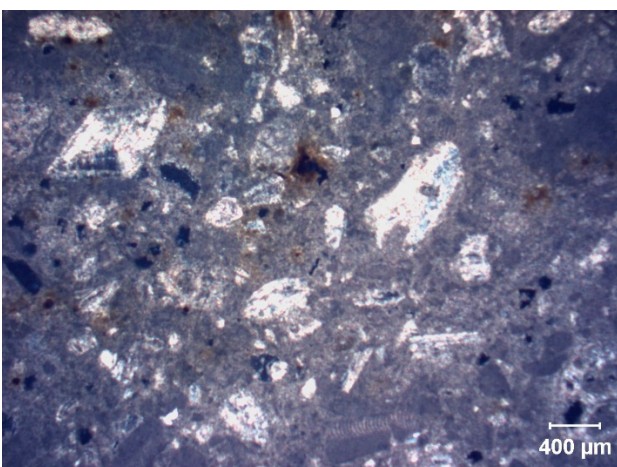

**Figure 10.** Photomicrographs of the tower wall sample (Polarized Optical Microscope with crossed nicols). Biomicrite limestone.

### 3.2.2. Facade Chapels Stone (s. XV)

It corresponds to a white limestone similar to the rock used in the tower (fine-grained biocalcarenite, biomicrite) (Figure 11). The fossils are foraminifera, mainly globigerinids, some fragments of bryozoans and mollusks. Small discontinuous venules of sparite are observed. Terrigenous content is low, corresponding mainly to quartz, opaque minerals and iron oxides. Other accessory minerals observed under SEM were apatite, zircon, ilmenite and titanite. According to XRD analysis, the mineralogical composition of this stone consisted of 96% calcite and 4% quartz.

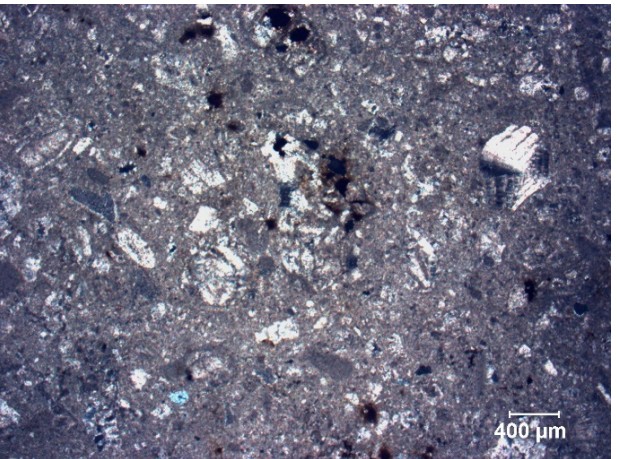

**Figure 11.** Photomicrographs of the central wall façade sample (Polarized Optical Microscope with crossed nicols). Biomicrite limestone.

### 3.2.3. Proto-Renaissance Doorway Stone (s. XVI)

According to the observation carried out at POM, it is a fossiliferous limestone with high matrix content (biomicrite) and a certain reef character (colonies of red algae and

bryozoans). It has abundant intraparticle porosity mainly associated with bryozoan components (Figure 12). Pores appear partially filled with a druse calcite cement (sparite). According to XRD analysis, the mineralogical composition of this rock consisted of 94% calcite, 3% quartz and 3% microcline.

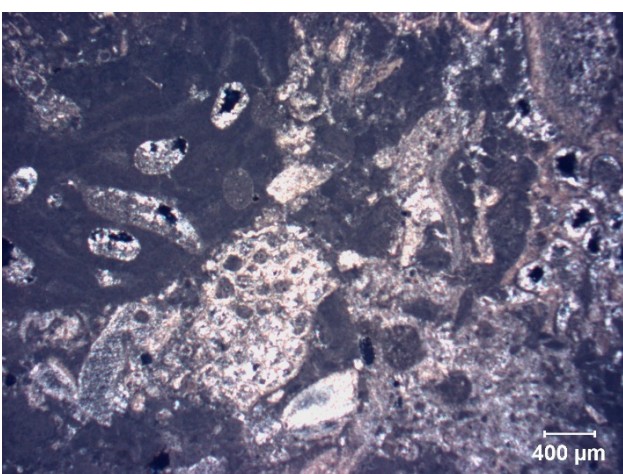

**Figure 12.** Photomicrographs Proto-Renaissance doorway sample (Polarized Optical Microscope with crossed nicols). Fossiliferous limestone.

### 3.2.4. Top Wall Façade Stone (s. XVIII)

This rock corresponds to a white limestone with greyish and reddish chromatic variations. Terrigenous content is high (around 10%), corresponding mainly to quartz, opaque minerals and glauconite. The fossiliferous content corresponds to bryozoans, foraminifera and limeclasts (Figure 13). According to XRD analysis, the mineralogical composition of this stone consisted of 94% calcite and 6% quartz.

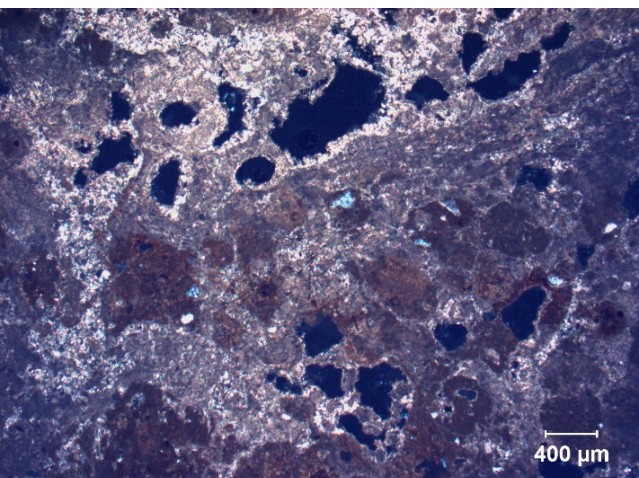

**Figure 13.** Photomicrographs at top wall façade sample (Polarized Optical Microscope with crossed nicols). Fossiliferous limestone.

### 4. Discussion

The documentary data researched indicate that the present church was built from at least the 15th century until the 18th century. The northeastern façade reflects this evolution over time, showing ashlar masonry interlinked over more than four centuries, which is why it has been taken as the object of study. It has been analysed and connected to the data obtained at documentary level and the geometric data obtained accurately (±5 mm of error in its total set) [36–38], with the data obtained from the lithological studies of its masonry.

The oldest documents that have been found date from the end of the 15th century, corresponding to the completion of the work on the interior of the present-day church, since in 1496 mention is made of finishing work on the side chapels. In this first stage, Gothic construction systems and Almohad measures typical of the 12th and 13th centuries were used. This could indicate that the space in which the present church was built was conditioned by the pre-existence of the previous mosque, partially adopting its measurements. It would also explain the existence of part of an old tower with dimensions that correspond to the use of measurements based on ma'mūnis cubits and the existence of traces of rammed earth on the upper part of the walls enclosing the main nave.

In this first Gothic stage of the present church, the façade enclosing the chapels was built, attached to an old tower. The reference plane taken in the point cloud study corresponds to the wall built in the 15th century.

From the analysis of the geometric anomalies obtained using 3D LiDAR, it can be seen that the plane that forms the façade of the Gothic chapels is mostly green in colour, which belongs to the plane of the façade, but there are areas that appear in black, indicating that they are located outside the range of the study carried out, which goes from (+30 mm) to (−30 mm). Most likely, the deformations were produced by the thrusts of the interior vaults.

The reference plane (the façade of the chapels) does not coincide exactly with the plane of the tower masonry, as the tower is slightly rotated (+30 mm). There is an obvious interface between the original tower and the Gothic wall. In Figure 9a–c, there is a very marked vertical strip coinciding with this construction joint between elements.

The fact that some of the stonemason marks on the Gothic façade coincide with those of other 14th and 15th century buildings in the province of Alicante could corroborate the construction period of this part of the church of Biar, as this masonry was built with regular ashlars of equal height in courses and do not appear to have been recycled.

The row of large ashlars, located under the one indicated as "5" in Figure 9c and which is highly degraded, must have finished off the Late Gothic façade completed in the 15th century. Today, an additional storey and half of the roof in this area rest on this façade, increasing the compressive load on the lower part of the wall and serving to increase the vertical component of the thrusts transmitted by the interior vaults on the façade. For more than two centuries, the intermediate zone (indicated as "6" in Figure 9c) was supporting higher horizontal thrusts, which deformed this zone outwards, as can be seen in the coloured graphs of the point cloud. These deformations are very small and would not have been identified by simple observation in situ. In the interior zone of the church where the symmetrical thrusts are transmitted, no deformations are observed in the point cloud. This is due to the fact that in this area there are more vertical loads transmitted by the upper wall that closes the central nave and collects the loads transmitted by the central stone vaults. These vertical forces minimise the horizontal thrusts transmitted by the vaults on the side chapels, and the resulting final force has a more vertical direction.

The Gothic rose window, which is currently partitioned and in the interior of the church, must have originally been located on the outside. This supports the hypothesis established in this work, that from the end of the 15th century until the 18th century there were no constructions over the side chapels.

At the beginning of the 16th century, the façade was completed by attaching it to the old tower. The shape of the cornices of the doorway probably made it necessary to dismantle several ashlars of the old tower's masonry, which remained outside the plane of the tower, as can be seen in the cloud of dots with colour gradient (Figure 9).

The four lithologies characterised are generally similar, being classified as "Biosparitic limestone with fossils", although they show certain differences between them. The stone of the old tower and the Gothic façade of the chapels are very similar, which would indicate that they are close in time of execution. The lithology of the doorway, where red algae appear, is slightly different from the previous ones. The lithology of the Baroque part where there is a higher percentage of terrigenous is also somewhat different.

Regarding the degradations, as we have seen in Section 3, the materials used in the construction of this façade are very porous and mainly of a limestone nature. The lower part of the wall shows multiple areas of red or black colour in the study of colour gradients, depending on the distance from the reference plane. This indicates distances greater than 3 cm from the reference plane. In Figure 9c, the area affected by the effects of capillary rising damp has been indicated as "2".

According to several studies on the damage caused by rising damp in the presence of salts [39,40], porous stone materials are particularly damaged, with a reduction in mechanical strength and an increase in porosity.

The crystallisation of salts accentuates the disintegration of the grains, as indicated by numerous authors [41,42], causing a continuous sandblasting over time in the crystallisation zone due to the effect of the pressures produced by the resulting crystals; salts also have a great hygroscopic capacity, being able to absorb water vapour and provoking in the materials that contain them the capacity to attract more water through an osmotic effect.

After rainy days, the presence of saline efflorescence has been observed, coinciding with the most degraded areas on the surface. In addition, the effect of rain with the presence of wind increases the effect of degradation on stone materials [43,44].

## 5. Conclusions

The current church of Nuestra Señora de la Asunción de Biar was begun in the 15th century in the area previously occupied by a mosque and a tower of which an elevation has been preserved. The northwest façade reflects the different construction stages in which the building was built until its completion in the 18th century.

The results obtained after the detailed analysis of the point cloud show that the method used in this work, with the study of deformations in gradient of colour with respect to a reference plane, has served to detect construction anomalies (such as small plane twists) and to corroborate the constructive evolutions of the church obtained from the documentary sources investigated. In addition, it has been possible to detect and quantify degradations and even small movements that would go unnoticed with the naked eye.

As can be deduced from the documentary studies, the observation in situ, the study of dimensions and the data obtained with the point cloud, the oldest area of the main façade corresponds to part of an old tower. The proportions of the masonry and its thickness correspond to cubits ma'munis, a measure used in the Almohad period, placing the construction of this part of the church in the AH 12th–13th centuries.

At the end of the 15th century, the side chapels to the main nave were built following Gothic construction systems, with stone vaults and pointed arches. The façade was annexed to the northeast of the existing medieval tower. These side chapels were not supposed to have a raised body, as is the case today. The height of the northeast façade at this stage was 650 cm, and the roof of the side chapels rested on the vaults. The central nave was to have more light thanks to a rose window that is now partitioned off and is located inside the chapter house.

At the beginning of the 16th century, the Proto-Renaissance doorway was attached to the other side of the old tower. This element of great architectural value was completed in 1522.

Later, in the 18th century, the current bell tower was built. An upper floor was also built on top of the existing chapels, which are accessed from the stairs of the Baroque tower. The façade was enlarged by adding a row of ashlars, followed by a cornice, and above this a façade of similar thickness to the lower part. This area tried to follow the same plan as the lower part, but as we have seen, it was slightly rotated, probably because the cornice did not allow them to have good plumb lines for reference during the works.

The characterization of the stones used in the construction of the different walls does not provide conclusive data in itself, since the lithologies are similar, with the exception of the one used on the Proto-Renaissance doorway, as it contains red algae. All the samples studied are biosparitic limestone with fossils.

The cloud of dots with gradient of colour, according to the distance from the reference plane, shows that erosion has occurred in the area affected by the runoff of rainwater from the old gargoyles. On the other hand, capillary rising damp has caused loss of material both in the ashlars and in the grouting mortar in the lower area, which corresponds to the plinth of the entire façade.

The row of ashlars where the first construction stage of the Late Gothic façade must have been finished off was unprotected against water seepage in its upper part for approximately two centuries, which is why it is more deteriorated than the surrounding rows.

The digitisation of heritage is essential for its correct understanding and its conservation over time. This work demonstrates the need to interrelate documentary and material data with the precise geometry achieved through digitisation of the built heritage. The analysis of the digitisation can also serve to complement the stratigraphic reading of the walls that may be carried out in the future by experts in architectural archaeology.

In addition, the data obtained from the geometric recording of this monument can serve as a starting point for the future control of possible deformations and degradations.

**Author Contributions:** Conceptualization, J.A.H.-T. and Y.S.-B.; methodology, J.A.H.-T. and Y.S.-B.; software, J.A.H.-T.; validation, J.A.H.-T., Y.S.-B. and P.S.-G.; Formal analysis, J.A.H.-T., P.S.-G. and Y.S.-B.; investigation, J.A.H.-T. and Y.S.-B.; resources, J.A.H.-T., Y.S.-B. and P.S.-G.; data curation, J.A.H.-T.; writing—original draft preparation, J.A.H.-T., Y.S.-B. and P.S.-G.; writing—review and editing, J.A.H.-T., Y.S.-B. and P.S.-G.; visualization, J.A.H.-T.; supervision, J.A.H.-T., Y.S.-B. and P.S.-G. All authors have read and agreed to the published version of the manuscript.

**Funding:** This research received no external funding.

**Data Availability Statement:** Data are contained within the article.

**Acknowledgments:** This work is related to the R&D&I project Challenges Research "Innovation in preventive response to directional extreme hydro-meteorological events on Cultural Heritage" subsidized by the Ministry of Science and Innovation of Spain.

**Conflicts of Interest:** The authors declare no conflicts of interest.

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
