# Peer review of "Use of LiDAR Technology for the Study and Analysis of Construction Phases and Deformations in the Gothic Church of Biar (Spain)"

_heritage, doi:10.3390/heritage7010006_

Round 1

Reviewer 1 Report

Comments and Suggestions for Authors

The work presents a work methodology of great value for the analysis of other monumental constructions. The incorporation of metrological and geometric studies is of great interest. These analyzes are usually ignored in studies of monumental architecture, but they provide valuable information about the time of construction and possible authorship. It is a very complete and coherent work that addresses analyzes aimed at very diverse objectives: from geometric, metrological and glyptographic analysis aimed at corroborating the evolution of the building, such as analyzes of the stone and deformations of the walls, aimed at detecting its stability. Particularly thorough is the analysis of the results obtained with the terrestrial laser scanner intended to verify the deformations suffered by the studied façade.

I raise two errors that must be modified:

In the summary the word Lidar is written on one occasion with lowercase letters. In the rest of the text it is always written in capital letters.

In number 4 of the references the surname Albero appears with the initial in lower case.

Author Response

I am attaching a response document to your comments and observations.
Thank you so much.

Reviewer 2 Report

Comments and Suggestions for Authors

The paper is very interesting, but lacking in some parts:

- no true graphic geometric analysis is reported. 

- line 187: specify how the data was interconnected.

- line 161: better to say " 80 stations point were carried out for the survey of...".

- line 141: no "Point clouds  in orthometric formats", but "orthophotos obtained from the 3D cloud point".

- there are no images on the archives documents.

- the caption of figure 2 is in Spanish.

- the bibliography needs to be updated.

Author Response

(The authors gave the same response as above.)

Reviewer 3 Report

Comments and Suggestions for Authors

name of the technology should be written like LiDAR - Light Detection And Ranging, so please correct all occurrences of the name - line 15 - Lidar, line 11 - LIDAR and so on

line 15 - LiDAR technology rather than LiDAR techniques - please correct in whole document

Key words - please add: LiDAR

line 30 - acronym BIC should be developed

line 33 - figures should be directly after the reference in the text - not 2 pages later

please don't cite the reference in blocks - 6-12 or 1-5, 7-12, it is better to summarize every publication in at least one sentence

line 88 why northeast is marked yellow?

line 143 - please use more engineering expressions instead of "high precision" - please add values

line 163 - what kind of mode of GPS was used - DGPS, SPP, RTK/RTN?

paragraph about TLS measurements 156-163 should be completed - localization of positions, registration errors, density of positioned data. Registration errors between scan positions are the crucial information in the aspects of plane based deformation. BLK too

line 170 - 3 to the superscript

line 182-185 - please add the numbers

line 269 - what it means "outside the study range"? more than +/- 30 mm?

fig 8 - presenting different values in the same colours is misleading - it would be better to present one larger figure divided into more compartments/colours. BTW 8d does not correspond to a, b and c - the plane from a/b/c drawn over the portal area would give differences in cm 40-60cm at least, and on c the area is in blue i.e. approx -30mm - please cut the fig. d. from the right.

 Typo " Figure 8. a, b y c)"

line 334 - in the publication the usual impersonal form is used, not "we"

line 335 - why the +/- 5mm value? on what basis? from registration report? from control measurement?

line 354 - The article does not provide arguments to support such a conclusion.

on the basis of which cloud were the analyses in fig. 8 made from BLK or from C10? from a single stand or from a combined cloud or from an integrated one?

42 literature items on 16 pages of text is not much. it will be good to add some refs for example:

A.https://fis.uni-bamberg.de/server/api/core/bitstreams/22e94e51-e30a-4586-a10d-28db4b4cfce0/content

B. https://isprs-archives.copernicus.org/articles/XLII-1-W2/61/2019/isprs-archives-XLII-1-W2-61-2019.pdf

C. https://iopscience.iop.org/article/10.1088/1755-1315/95/3/032007/pdf

Author Response

(The authors gave the same response as above.)

Round 2

Reviewer 3 Report

Comments and Suggestions for Authors

Thank you for applying my comments.

One issue left - please add blue response information into text of the manuscript.

on the basis of which cloud were the analyses in fig. 8 made from BLK or from C10? from a single stand or from a combined cloud or from an integrated one?

All the analysis has been carried out through clouds obtained from a single Terrestrial laser scanner, the C10 model. From the combined cloud of all the stations point, the error of the whole of this station point is 4 mm and a total cloud-to-cloud error is 3 mm.

Author Response

Comments are provided to improve the text by adding paragraphs according to the reviewer's recommendations.
We appreciate your valuable contributions.
